# Synthesis of piperidines and pyridine from furfural over a surface single-atom alloy Ru₁Co_NP catalyst

Haifeng Qi [1,2,5], Yurou Li[3,5], Zhitong Zhou[1], Yueqiang Cao [3], Fei Liu[1], Weixiang Guan[1], Leilei Zhang [1], Xiaoyan Liu [1], Lin Li[1], Yang Su[1], Kathrin Junge [2], Xuezhi Duan [3] ✉, Matthias Beller [2] ✉, Aiqin Wang [1,4] ✉ & Tao Zhang [1,4]

The sustainable production of value-added N-heterocycles from available biomass allows to reduce the reliance on fossil resources and creates possibilities for economically and ecologically improved synthesis of fine and bulk chemicals. Herein, we present a unique Ru₁Co_NP/HAP surface single-atom alloy (SSAA) catalyst, which enables a new type of transformation from the bio-based platform chemical furfural to give N-heterocyclic piperidine. In the presence of NH₃ and H₂, the desired product is formed under mild conditions with a yield up to 93%. Kinetic studies show that the formation of piperidine proceeds via a series of reaction steps. Initially, in this cascade process, furfural amination to furfurylamine takes place, followed by hydrogenation to tetrahydrofurfurylamine (THFAM) and then ring rearrangement to piperidine. DFT calculations suggest that the Ru₁Co_NP SSAA structure facilitates the direct ring opening of THFAM resulting in 5-amino-1-pentanol which is quickly converted to piperidine. The value of the presented catalytic strategy is highlighted by the synthesis of an actual drug, alkylated piperidines, and pyridine.

Biomass, as the only renewable organic carbon resource in nature, will be a key feedstock for a circular chemical industry and provide the basis for the production of most value-added chemicals in the future[1-5]. In this respect, significant progress has been achieved in the transformation of lignocellulose to oxygen-containing compounds via the selective cleavage of C-C and C-O bonds over supported metal catalysts in the past decade[6-11]. Compared to oxygenated chemicals, N-containing compounds usually have higher values and are broadly applied in the production of pharmaceuticals and agrochemicals[12,13]. In fact, more than 75% of top 200 selling drugs contain amine/nitrogen moieties[12,13]. Furthermore, there is a growing demand for new bioactive compounds and improved personal care products, which also

require new amine building blocks. Thus, it is not surprising that growing research attention is paid towards the sustainable production of such chemicals from biomass[14-21]. So far, various aliphatic amines have been synthesized via reductive amination of biomass-derived aldehydes/ketones/alcohols[15,21-23]. One important biomass-derived building block is furfural, which is cheap (1.0–1.2 € kg⁻¹) and readily available on a large scale from biomass (>200 kT per year)[24]. State-of-the-art routes for furfural amination focused on the reaction between the aldehyde group of furfural and organic amines or NH₃ in the presence of H₂ over Ru-, Rh-, Pd-, Co-, or Ni-based catalysts, merely leading to the corresponding primary and secondary amines (Fig. 1A)[15,24-29]. Compared to such amines, N-heterocycles (e.g., piperidine, pyridine,

[1]CAS Key Laboratory of Science and Technology on Applied Catalysis, Dalian Institute of Chemical Physics, Chinese Academy of Sciences, Dalian 116023, China. [2]Leibniz-Institut für Katalyse, Albert-Einstein-Straße 29a, Rostock 18059, Germany. [3]State Key Laboratory of Chemical Engineering, East China University of Science and Technology, Shanghai 200237, China. [4]State Key Laboratory of Catalysis, Dalian Institute of Chemical Physics, Chinese Academy of Sciences, Dalian 116023, China. [5]These authors contributed equally: Haifeng Qi, Yurou Li. ✉e-mail: xzduan@ecust.edu.cn; matthias.beller@catalysis.de; aqwang@dicp.ac.cn

**A** Previous works for the reductive amination of furfural ──────────

- **Furanic amines**
- **Pyrrole synthesis**

**B** Piperidine-based pharmaceuticals ──────────

**Artane**          **Marcaine**          **Aplace**          **Sublimaze**

**C** The reaction network for the transformation of furfural to piperidine and **this work** ──────

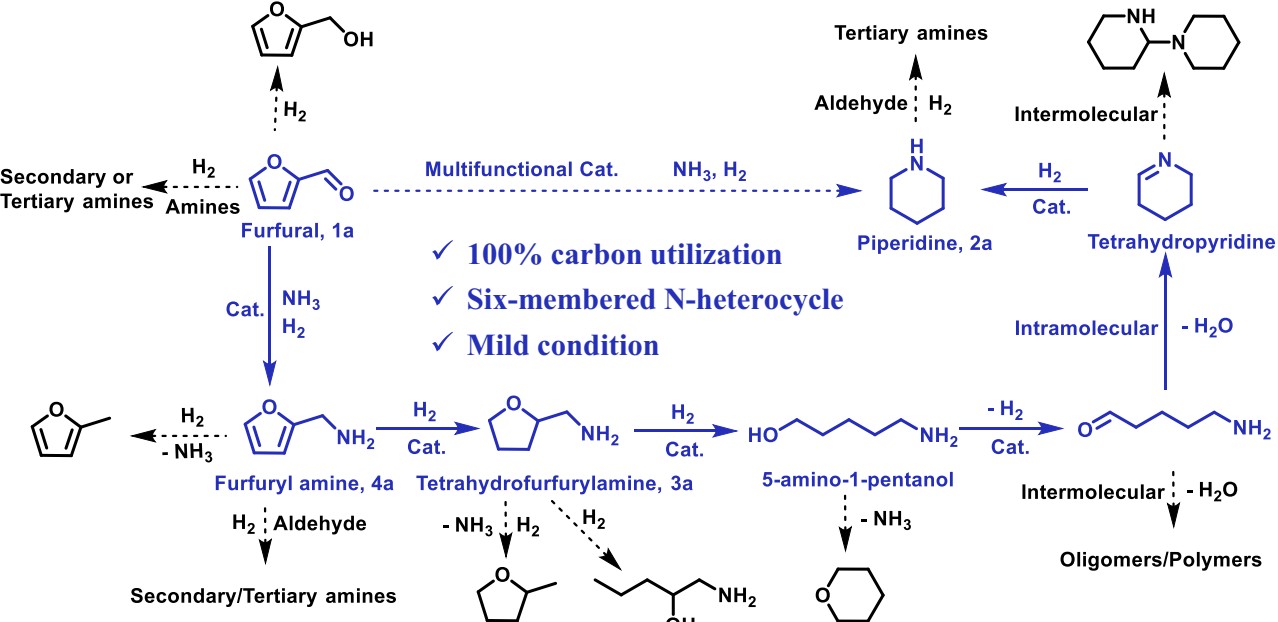

✓ **100% carbon utilization**
✓ **Six-membered N-heterocycle**
✓ **Mild condition**

**Fig. 1 | Previous and present works for the amination of furfural. A** Furanic amines and pyrrole synthesis from furfural in previous works; (**B**) examples of piperidine-based pharmaceuticals; and (**C**) proposed reaction network for the transformation of furfural to piperidine in this work.

pyrrole, etc.) have been rarely reported to be produced from biomass due to the lack of N-heterocyclic rings in most of biomass-derived platform molecules. Apart from pyrroles, which can be synthesized from biomass-derived furans, other N-heterocycles have been scarcely investigated. For example, Cao and co-workers reported the synthesis of N-substituted pyrroles via direct condensation of biomass-derived 2,5-dimethylfuran with mainly anilines over H-Y zeolite catalyst[30]. Later, Yan and co-workers developed a route to access pyrrole from furfural via tandem decarbonylation-amination reactions over Pd@S-1 and H-beta zeolite catalyst system (Fig. 1A)[22]. While a pyrrole selectivity of 75% was achieved, the reaction was operated at 460 °C and the decarbonylation step undoubtedly reduced the atom economy by discarding aldehyde group of furfural. Obviously, current methodologies to construct bio-based N-heterocycles rely critically on the use of

structurally similar biomass-derived O-heterocycles (furan derivatives), which limits the possibilities to create other N-heterocycles, e.g., only pyrroles are produced from either furan or furfural derivatives.

Piperidine is a frequently encountered molecular fragment in thousands of pharmaceutically active compounds (e.g., Artane, Marcaine, Aplace, and Sublimaze, Fig. 1B), which is also highlighted in the book "Piperidine-Based Drug Discovery"[31]. Motivated by importance of this class of compounds, we envisioned a direct transformation of furfural (FAL) with ammonia to provide piperidine (Fig. 1C). As reported by previous works[32–35] and shown in Fig. 1C, this reaction cascade starts with the reductive amination of FAL, followed by hydrogenative ring cleavage, dehydrogenation, intramolecular reductive amination, and final hydrogenation to produce piperidine. Obviously, it is difficult to achieve a selective process in this highly

**Table 1 | Reductive amination of furfural over supported $Ru_xCo_y$ materials[a]**

| Entry | Catalysts[b] | Metal Loading (wt%) | | Yield (%)[d] | | | Production rate ($mol_{2a} \cdot mol_{Ru}^{-1} \cdot h^{-1}$) |
|---|---|---|---|---|---|---|---|
| | | Ru | Co | 2a | 3a | 4a | |
| 1 | Co/HAP | – | 5.0 | n.d. | n.d. | 59 | – |
| 2 | Ru/HAP | 0.50 | – | n.d. | 88 | n.d. | – |
| 3 | $Ru_{20}Co_1$/HAP | 5.0 | 0.15 | n.d. | 89 | n.d. | – |
| 4 | $Ru_{10}Co_{10}$/HAP | 4.10 | 2.40 | n.d. | 91 | n.d. | – |
| 5 | $Ru_5Co_{10}$/HAP | 2.0 | 2.6 | n.d. | 90 | n.d. | – |
| 6 | $Ru_{11.6}Co_{20}$/HAP | 5.0 | 5.0 | n.d. | 87 | n.d. | – |
| 7 | $Ru_{4.6}Co_{20}$/HAP | 2.0 | 5.0 | 2 | 89 | n.d. | – |
| 8 | $Ru_2Co_{20}$/HAP | 0.85 | 4.9 | 72 | 16 | n.d. | 4.3 |
| 9 | $Ru_1Co_{20}$/HAP | 0.43 | 5.0 | 91 | n.d. | n.d. | 11.6 |
| 10 | $Ru_1Co_{40}$/HAP | 0.21 | 4.9 | 93 | n.d. | n.d. | 22.6 |
| 11 | $Ru_1Co_{80}$/HAP | 0.10 | 5.0 | 92 | n.d. | n.d. | 44.7 |
| 12 | $Ru_1Co_{160}$/HAP | 0.05 | 5.0 | 97 | n.d. | n.d. | 94.2 |
| 13 | $Ru_1Co_{20}$/$Al_2O_3$ | 0.40 | 5.0 | n.d. | 54 | n.d. | – |
| 14 | $Ru_1Co_{20}$/MgO | 0.40 | 5.0 | n.d. | 63 | n.d. | – |
| 15 | $Ru_1Co_{20}$/$TiO_2$ | 0.40 | 5.0 | n.d. | 86 | n.d. | – |
| 16 | $Ru_1Co_{20}$/$ZrO_2$ | 0.40 | 5.0 | n.d. | 87 | n.d. | – |
| 17[c] | 0.5Ru/HAP + 5Co/HAP | 0.5 | 5.0 | n.d. | 84 | n.d. | – |

[a]Reaction conditions: 50 mg catalyst, 0.5 mmol furfural, 5 g p-xylene, 0.5 MPa $NH_3$, 1 MPa $H_2$, 100 °C for 6 h and 180 °C for 14 h, dodecane as internal standard; [b]the subscripts of Co and Ru represent the Co/Ru molar ratio; [c]Physical mixture, 50 mg 0.5Ru/HAP and 50 mg 5Co/HAP; [d]The conversions of furfural were all >99%, and other products were oligomers and polymers.

complicated reaction network and especially the selective C-O cleavage of the furan ring under the condition of reductive amination is a most critical reaction step. Thus, key to the success for the realization of this new strategy lies in the delicate design of a multifunctional catalyst.

Herein, we report a multifunctional surface single-atom alloy (SSAA) $Ru_1Co_{NP}$/HAP catalyst, which shows unique activity for the envisioned one-pot conversion of furfural to piperidine, affording the desired product in excellent yield (93%) under mild conditions. Thus, value-added drugs such as Artane, alkylated piperidines, and pyridine as well as its derivatives can be obtained from renewables by further valorisation of piperidine. Characterizations in conjunction with theoretical studies show that the support (hydroxyapatite, HAP) facilitates the formation of atomically dispersed Ru species on the surface of Co nanoparticles (NPs), and the resultant $Ru_1Co_{NP}$ SSAA structure plays a key role in hydrogenative ring rearrangement of the intermediate tetrahydrofurfurylamine (THFAM) towards piperidine.

## Results and discussion
### Catalyst design and performance
Our approach to develop an active catalyst for the desired transformation was based on the assumption that both Co- and Ru-based materials showed good activity for the reductive amination of biomass-derived aldehydes[21,27]. It is also known that bimetallic alloy catalysts outperform the individual ones in C-O cleavage and amination reactions[36,37]. Consequently, we prepared a series of supported $Ru_xCo_y$ materials and investigated their catalytic performances for the direct amination of FAL. To achieve high selectivity in the initial reductive amination step and avoid unwanted side-products, the reaction was operated at 100 °C for 6 h, then the temperature was

increased to 180 °C for 14 h to promote the hydrogenative C-O cleavage and ring rearrangement. As shown in Table 1, the known Co/HAP material afforded furfuryl amine (FAM, 4a) as the major product in moderate yield (59%), while the Ru/HAP gave rise to THFAM (3a) 88% yield (Table 1, entries 1-2), which is in accordance with a higher hydrogenation activity of Ru compared to Co[29]. Similar to the monometallic Ru/HAP, a series of $Ru_xCo_y$/HAP materials gave THFAM as the major product, when the Co/Ru atomic ratio is smaller than 10/1 (Table 1, entries 3-7), which implied that Ru is most likely enriched on the surface in these bimetallic catalysts forming an extended Ru surface similar to that of monometallic Ru/HAP. Surprisingly, a drastic change in the catalytic behaviour occurred, when the Co/Ru ratio was equal to or greater than 10/1, which resulted in the formation of piperidine as the primary product (Table 1, entries 8–10). To the best of knowledge, such transformation - selective formation of six-membered N-heterocycles via reductive amination of furfural - has not been reported, yet. Notably, the yield of piperidine increased with lowering the Ru content and achieved 93% at a Co/Ru ratio of 40/1. From an academic and practical point of view, the improved activity with decreasing Ru amount is remarkable and only at very dilute concentrations of Ru piperidine formation occurred. Consequently, when the piperidine production rates were calculated based on Ru (assuming Ru-Co as the active site), an increase with the Co/Ru ratio: $Ru_1Co_{40}$/HAP (22.6 $mol_{2a} \cdot mol_{Ru}^{-1} \cdot h^{-1}$) > $Ru_1Co_{20}$/HAP (11.6 $mol_{2a} \cdot mol_{Ru}^{-1} \cdot h^{-1}$) > $Ru_1Co_{10}$/HAP (4.3 $mol_{2a} \cdot mol_{Ru}^{-1} \cdot h^{-1}$) was observed. Considering the highly diluted concentration of Ru in the $Ru_1Co_{40}$/HAP catalyst, we assumed that Ru species are sufficiently isolated by Co atoms and form surface single-atom $Ru_1Co_{NP}$ alloy structure which is responsible for the formation of

piperidine. In order to further prove this hypothesis, another two $Ru_1Co_{80}$/HAP and $Ru_1Co_{160}$/HAP catalysts with much lower Ru content were prepared, which gave piperidine as the product with relatively higher production rates (44.7 $mol_{2a}·mol_{Ru}^{-1}·h^{-1}$ and 94.2 $mol_{2a}·mol_{Ru}^{-1}·h^{-1}$, respectively, Table 1, entries 11-12).

In addition to the Co/Ru ratio, the HAP support also played an important role in this catalytic system. As shown in Table 1, other applied supports, whether acidic or basic ($Al_2O_3$ and MgO), reducible or non-reducible ($TiO_2$ and $ZrO_2$), all showed dominant formation of THFAM (Table 1, entries 13-16), even when the optimized Co/Ru ratio of 20/1 or higher reaction temperature was applied (Supplementary Tables 1-2). Further, the same catalytic performance of physically mixed 0.5Ru/HAP and 5Co/HAP with individual Ru/HAP indicated the specific role of CoRu alloy for the piperidine formation (Table 1, entry 17).

To proof the postulated reaction pathway for the formation of piperidine from furfural (Figs. 1C and 2c), kinetic profiles of two standard reactions in the presence of $Ru_1Co_{20}$/HAP and $Ru_{10}Co_{10}$/HAP were done. Utilizing the $Ru_1Co_{20}$/HAP catalyst (Fig. 2a), Schiff base (i.e., secondary imine) 5a is formed at the beginning of the reaction, and then rapidly consumed to produce FAM 4a. Subsequent hydrogenation led to THFAM 3a. Increasing the reaction temperature to 180 °C, the yield of 3a reaches a maximum of 62% and then decreases because of the formation of piperidine 2a. The yield of 2a increases monotonically with the reaction time until it levels off (90%) after 16 h. The shown kinetic profile strongly suggests that THFAM 3a is the key intermediate to produce piperidine 2a. Compared to $Ru_1Co_{20}$/HAP, the $Ru_{10}Co_{10}$/HAP catalyst exhibits a similar kinetic profile for both the formation of 4a and 3a, but 3a is not further converted and becomes the final product even after extended time (Fig. 2b). A series of control experiments with 3a as the substrate reveal the same trend as shown in Table 1. In fact, among these tested $Ru_xCo_y$ catalysts, only the $Ru_xCo_y$/HAP catalysts with a low-Ru content (Co/Ru ≥ 10) allow for almost quantitative conversion of 3a to 2a, whereas the other catalysts are inactive (see Supplementary Table 2). Based on these results, the key step in this transformation is the reaction of 3a to 2a (Fig. 2c),

which is only enabled by $Ru_xCo_y$/HAP catalysts with low Ru content (Co/Ru ≥ 10).

## Structural characterizations, mechanism, and DFT calculations

In order to understand the unique selectivity of the active catalysts to give piperidine, $Ru_1Co_{20}$/HAP with relatively higher Ru content was chosen as a representative sample for further characterizations. The $H_2$-temperature-programmed reduction ($H_2$-TPR) profile of the calcined $Ru_1Co_{20}$/HAP (Supplementary Fig. 1) presents two major peaks at 122 °C and 284 °C, which are attributed to the reduction of $RuO_2$ and $Co_3O_4$, respectively[29,38,39]. In addition, a minor peak occurs at 165 °C, which is not observed for either monometallic Ru/HAP and Co/HAP samples or bimetallic $Ru_{10}Co_{10}$/HAP. Therefore, this peak is unique to the $Ru_1Co_{20}$/HAP sample and ascribed to the co-reduction of Ru and Co to form Ru-Co surface alloy structure. It is further evident that the presence of Ru promotes the reduction of Co due to spillover of dissociated hydrogen atoms[40,41]. In line with the $H_2$-TPR, the in-situ X-ray diffraction (XRD) patterns of the $Ru_1Co_{20}$/HAP under $H_2$ atmosphere show the appearance of a metallic Co peak and concurrent disappearance of the $Co_3O_4$ peak when the reduction temperature reaches 250 °C, which is 200 °C lower than Co/HAP (Supplementary Fig. 2). The absence of any reflections of Ru in the XRD pattern of reduced $Ru_1Co_{20}$/HAP suggests the high dispersion of Ru species due to the low concentration of Ru in this sample. For comparison, the reflection of metallic Ru is evident in $Ru_{10}Co_{10}$/HAP (Supplementary Fig. 3), indicating the formation of large Ru NPs. Figure 2a, b display high-angle annual dark-filed scanning transmission electron microscopy (HAADF-STEM) images of the $Ru_1Co_{20}$/HAP and $Ru_{10}Co_{10}$/HAP catalysts. One can see nonuniform particles ranging from several to tens of nanometres in both samples. Elemental mapping shows that these NPs are composed of both Co and Ru. In particular, the weak and highly dispersed signals of Ru are overlapped with dense and strong Co signals in $Ru_1Co_{20}$/HAP catalyst, which suggests that Ru is most likely atomically dispersed on the Co nanoparticles. In sharp contrast, the Ru signals in $Ru_{10}Co_{10}$/HAP are almost as dense as those of Co,

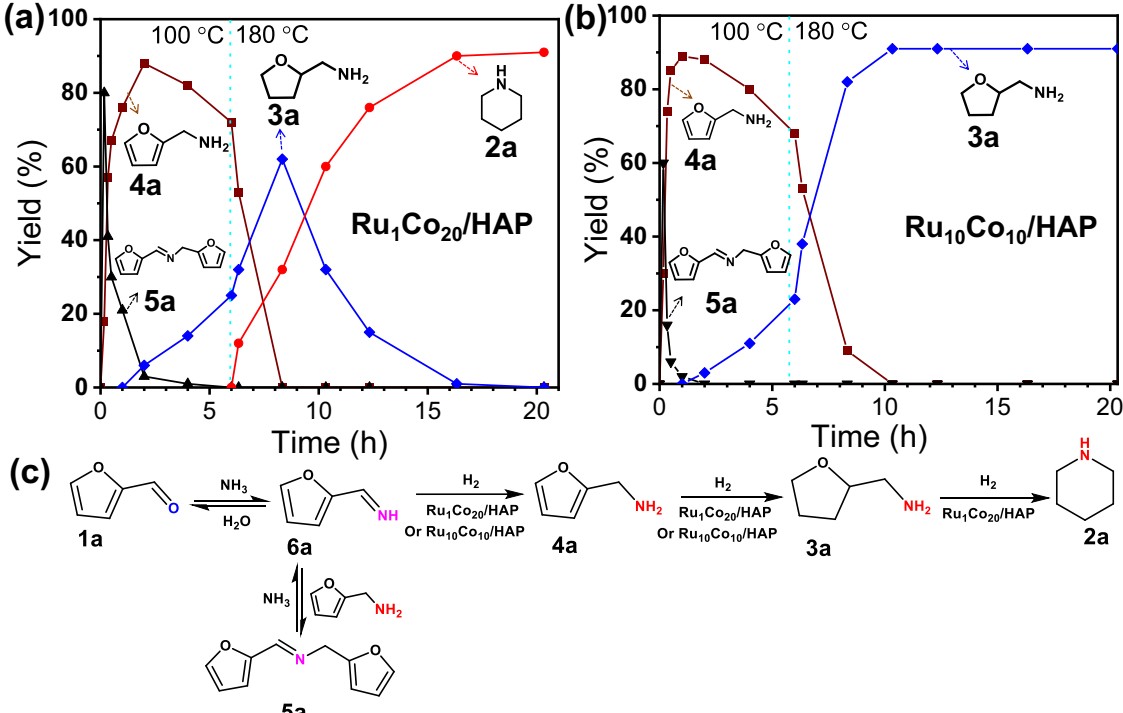

**Fig. 2 | Kinetic profiles.** Reductive amination of furfural over (**a**) $Ru_1Co_{20}$/HAP and (**b**) $Ru_{10}Co_{10}$/HAP catalysts, and proposed reaction pathways (**c**) Reaction conditions: 0.5 mmol furfural, 50 mg catalyst, 5 g *p*-xylene, 0.5 MPa $NH_3$, 1 MPa $H_2$, 100 °C, 6 h, subsequently (20 min), 180 °C, 14 h, dodecane as internal standard.

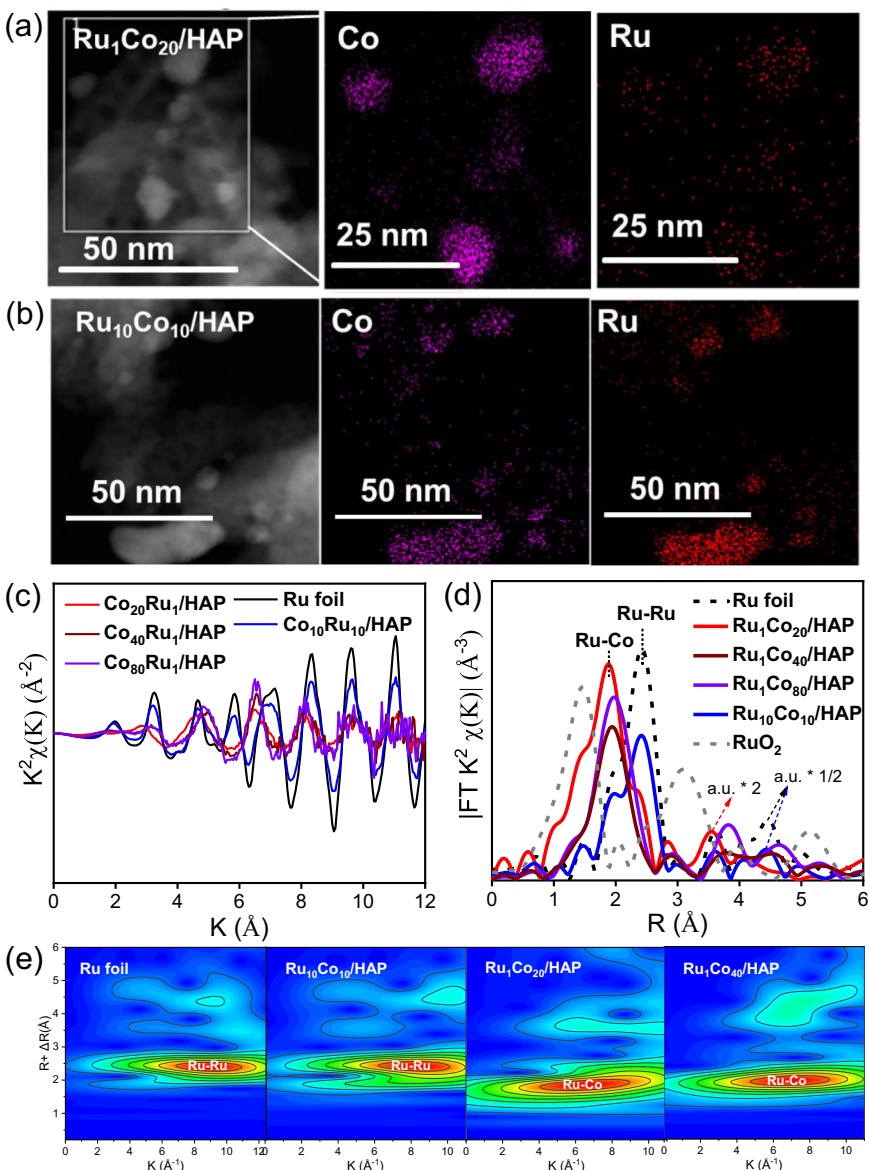

**Fig. 3 | Characterizations for Ru$_x$Co$_y$/HAP catalysts.** High-angle annual dark-filed scanning transmission electron microscopy images and elemental mapping of Co and Ru in Ru$_1$Co$_{20}$/HAP (**a**) and Co$_{10}$Ru$_{10}$/HAP (**b**); the $k^2$-weighted Fourier transform extended X-ray absorption fine structure spectra in $k$-space (**c**) and $r$-space (**d**), as well as wavelet transforms (**e**) of Ru$_1$Co$_{20}$/HAP, Ru$_1$Co$_{40}$/HAP, Ru$_1$Co$_{80}$/HAP, and Ru$_{10}$Co$_{10}$/HAP.

suggesting the formation of a Ru-Ru ensemble structure which might be also aggregated on the Co or Co-rich NPs.

Since the HAADF-STEM imaging could not resolve Ru single atoms in the presence of relatively large Co NPs, X-ray absorption spectra (XAS) at the Ru $K$-edge was conducted to provide further evidence about the local coordination of Ru atoms. Indeed, in the $k$-space of the extended X-ray absorption fine structure spectra (EXAFS), the Ru$_1$Co$_{20}$/HAP shows a quite different pattern from those of Ru foil and Ru$_{10}$Co$_{10}$/HAP (Fig. 3c), suggesting the formation of a Co-Ru alloy structure. The $k^2$-weighted Fourier transform of the EXAFS in $r$-space of the Ru$_1$Co$_{20}$/HAP (Fig. 3d) displays a major peak at 1.95 Å and a minor shoulder at 2.40 Å (not phase-corrected), which could be ascribed to Ru-Co and Ru-Ru contribution, respectively. To further resolve Ru-Ru and Ru-Co coordination, wavelet transform (WT) of Ru $K$-edge EXAFS oscillations was carried out owing to its more powerful resolutions in both $k$ and $r$ spaces (Fig. 3e)[42]. The same lobe at (2.4 Å, 8.8 Å$^{-1}$), which is associated with Ru-Ru contribution, is clearly observed from the WT contour plots of Ru foil standard and Ru$_{10}$Co$_{10}$/HAP sample, indicating a dominating Ru-Ru contribution in the Ru$_{10}$Co$_{10}$/HAP sample. In

contrast, the WT lobe of Ru$_1$Co$_{20}$/HAP sample shifts to a lower position in both $r$ and $k$ values (1.9 Å, 6.4 Å$^{-1}$), indicating a dominant Ru-Co coordination. Consistently, the best-fitted EXAFS result of the Ru$_1$Co$_{20}$/HAP sample reveals Ru-Co shell at 2.52 Å with coordination number (CN) of 6.9 and Ru-Ru shell at 2.59 Å with CN of 3.4 (Supplementary Fig. 4a and Supplementary Tables 3-4). The ratio of Ru-Co CN to Ru-Ru CN is 2.1, which is slightly lower than the surface Co/Ru ratio determined by XPS (3.2, Supplementary Table 5), but much lower than the bulk Co/Ru ratio determined by ICP (20, Supplementary Table 5), indicating the prevalence of surface Ru$_1$Co$_{NP}$ single-atom alloy structure along with few Ru-Ru small groups. Further evidence for these latter ensembles of Ru-Ru is provided by the shorter Ru-Ru distance compared to Ru foil (2.59 vs. 2.67 Å). In contrast to Ru$_1$Co$_{20}$/HAP catalyst, the Ru$_{10}$Co$_{10}$/HAP sample shows similar spectra to Ru foil in both $k$-space and $r$-space, and the best-fitted result presents Ru-Ru coordination at 2.66 Å with CN of 8.1 and Ru-Co coordination at 2.54 Å with CN of 1.1 (Supplementary Fig. 4d and Supplementary Tables 3-4). The ratio of Ru-Ru CN to Ru-Co CN reaches 7.4, which is significantly higher than what is expected from the chemical composition (1/1). Obviously,

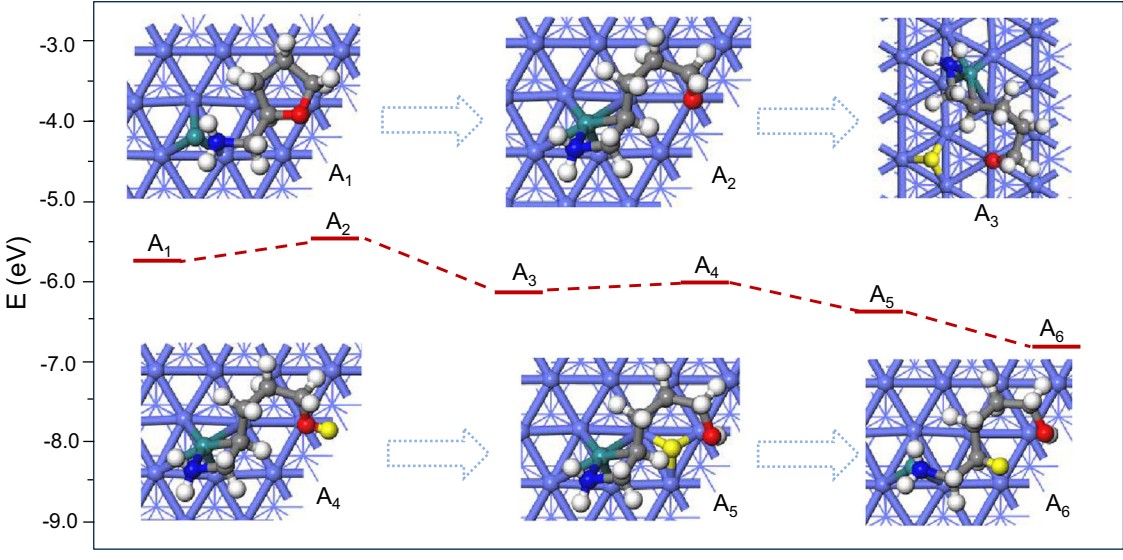

**Fig. 4 | DFT calculations.** The mechanistic investigation of THFAM to piperidine on $Ru_1$/Co (001) surface. $A_1 \rightarrow A_2$: the direct scission of the C-O bond near the -$NH_2$ group of THFAM on the $Ru_1$/Co surface; $A_2 \rightarrow A_3$: H atom from $H_2$ dissociation is loaded on the $Ru_1$/Co surface; $A_3 \rightarrow A_4$: the activated H binds with the O atom of the C-O bond; $A_4$-$A_6$: the following hydrogenation to the intermediate 5-amino-1-pentanol; the color of ball represents: green: Ru, purple: Co, blue: N, red: O, grey: C, white/yellow: H.

the Ru and Co in the $Ru_{10}Co_{10}$/HAP did not form uniform alloy structure; instead, phase segregation occurred with Ru-Ru ensembles enriched on the surface. These structural differences are the main reason why the catalytic performance of this latter sample resembled monometallic Ru/HAP.

Based on the above results, it can be reasonably postulated by comparing the different structures of $Ru_1Co_{20}$/HAP and $Ru_{10}Co_{10}$/HAP, that the Ru atoms in the $Ru_1Co_y$/HAP catalysts (y = 40, 80, and 160) are sufficiently isolated by the neighbouring Co atoms, thus providing a unique SSAA structure to enable the one-pot formation of piperidine from FAL, as illustrated in Supplementary Fig. 5. In order to prove our hypothesis, the $Ru_1Co_{40}$/HAP and $Ru_1Co_{80}$/HAP catalysts with relatively lower Ru content were further chosen for XAS characterizations. As shown in Fig. 3c, d, the different *k*-space patterns of the $Ru_1Co_{40}$/HAP and $Ru_1Co_{80}$/HAP with that of Ru foil as well as only one peak at ~1.98 Å (not phase-corrected) in *r*-space of these two catalysts, suggest the formation of sole Ru-Co coordination, which is also confirmed by wavelet transforms (Fig. 3e). The best-fitted EXAFS results of the $Ru_1Co_{40}$/HAP and $Ru_1Co_{80}$/HAP samples reveal Ru-Co shell at 2.46 Å with CN of 5.3 and Ru-Co shell at 2.48 Å with CN of 5.7, respectively (Supplementary Fig. 4b-c and Supplementary Table 3). The absence of Ru-Ru coordination clearly indicates the isolated Ru atoms anchored on Co nanoparticles in the low-Ru-loading $Ru_1Co_y$/HAP samples (y = 40, 80, and 160).

Then, the surface properties of the $Ru_1Co_{20}$/HAP catalyst are further probed by CO-adsorbed Fourier transform infrared spectroscopy (FT-IR). Both Ru/HAP and $Ru_{10}Co_{10}$/HAP show bands at around 2139 ~ 2100, 2081 ~ 2050, and 2003 ~ 1850 $cm^{-1}$, which could be assigned to the stretching vibrations of $Ru^{n+}$-$(CO)_x$, $Ru^0$ NP-CO, and $Ru_2^0$-$CO$[43], respectively, indicative of the predominance of the Ru surface. In contrast, no bands are detected for the $Ru_1Co_{20}$/HAP sample, which is probably due to the formation of the surface single-atom alloy $Ru_1Co_{NP}$ structure that suppresses CO adsorption, which is in agreement with earlier reports about SAA[44–46]. More interestingly, when comparing different supported $Ru_1Co_{20}$ catalysts, it is found that all the samples exhibit similar CO absorption bands compared to the pure Ru/HAP except for the HAP-supported one (Supplementary Fig. 6), which also suggests that the HAP support plays a unique role in

promoting the formation of a surface single-atom alloy $Ru_1Co_{NP}$ structure. Furthermore, in-situ THFAM-adsorbed IR spectroscopy was conducted to exclude the participation of HAP in the key transformation of intermediate THFAM to product piperidine. As seen in Supplementary Fig. 7, a strong THFAM-adsorbed IR signal on $Ru_1Co_{20}$/HAP sample compared to that of almost no signal on HAP alone, indicates $Ru_1Co_{20}$ single-atom alloy could efficiently adsorb or activate THFAM rather than support.

To understand the beneficial effect of the $Ru_1Co_{NP}$ SSAA structure for the catalytic transformation of THFAM to piperidine, we performed density functional theory (DFT) calculations using thermodynamically stable $Ru_1$/Co (001) to represent the $Ru_1Co_{NP}$/HAP catalyst[47]. The optimized THFAM adsorption configurations show that THFAM is strongly adsorbed on the $Ru_1$/Co (001) surface with the N atom of the amino group binding to the $Ru_1$ atom while the tetrahydrofuran ring binds to the Co surface, and the adsorption energy is −5.70 eV (Fig. 4 and Supplementary Fig. 8).

Then, direct scission of the C − O bond near the -$NH_2$ group of THFAM on the $Ru_1$/Co surface occurs favourably ($A_1 \rightarrow A_2$) with a low energy barrier (0.26 eV). The subsequent hydrogenation proceeds energetically downhill until the formation of 5-amino-1-pentanol ($A_2 \rightarrow A_3 \rightarrow A_4 \rightarrow A_5 \rightarrow A_6$). Finally, 5-amino-1-pentanol undergoes ring closure by formal dehydration leading to the formation of piperidine, shown in Supplementary Fig. 9. To further verify this reaction mechanism, we performed temperature-programmed desorption (TPD) of 5-amino-1-pentanol under $N_2$ in the presence of the $Ru_1Co_{20}$/HAP catalyst. In this experiment, piperidine is detected at 252 °C, while 5-amino-1-pentanol is desorbed at 350 °C (Supplementary Fig. 10). Obviously, the ring-closing of 5-amino-1-pentanol via dehydration proceeded before it was desorbed from the catalyst surface, that is the reason why we did not detect 5-amino-1-pentanol in the kinetic profiles (Fig. 2a), also confirmed by quickly transformation rate of 5-amino-1-pentanol to piperidine over $Ru_1Co_{20}$/HAP catalyst (Supplementary Table 6). This is in good agreement with the DFT calculations.

## Practical applications and valorization of piperidines

Apart from the high activity and unique selectivity displayed by the $Ru_1Co_{NP}$/HAP SSAA catalyst for the direct transformation of furfural

**A** Sustainable production of bulk chemical pyridine from furfural

**B** Renewable synthesis of value-added drug from furfural

**C** Synthesis of 2-substituted piperidines from furfural derivatives

**Fig. 5 | Valorization of bio-based piperidines. A** the renewable two-step synthesis of pyridine from furfural; (**B**) the synthetic process of Artane from furfural; and (**C**) the preparation of 2-substituted piperidines from different substrates of 5-substituted furfural via same catalytic system.

to piperidine, other aspects such as reusability and scale-up property are important for any practical application. Hence, five recycling tests under the standard conditions were performed by recovered reduction after each cycle, and the yields of piperidine maintained stable (Supplementary Fig. 11). Also, the gram-scale experiment was conducted smoothly with moderate piperidine yield under standard reaction conditions (72%, Supplementary Table 7).

In general, the use of bio-based piperidines permits a renewable synthesis of various existing fine and agrochemicals as well as pharmaceuticals *vide supra*. Exemplarily, as another important application of the novel Ru$_1$Co$_{20}$/HAP catalyst, the direct synthesis of bulk chemical pyridine could be realized. This one-pot reaction could be extended to produce such important N-heteroarene at 240 °C in 88% yield based on FAL simply by switching the H$_2$ gas to N$_2$ (Fig. 5A). Also, this is showcased by the synthesis of Artane which is an important drug for the treatment of Parkinson's syndrome[31]. Firstly, piperidine was extracted by aqueous hydrochloric acid giving pure piperidine hydrochloride (Fig. 5B, NMR, Supplementary Fig. 12), which can be used to treat bronchitis and emphysema[48]. Further reaction with acetophenone and paraformaldehyde under reflux conditions gave β-piperidinopropiophenone (NMR, Supplementary Fig. 13). In the final reaction step with cyclohexylmagnesium chloride the desired product is obtained in 72% overall yield based on furfural (Fig. 5B, NMR, Supplementary Fig. 14). Finally, few different 5-substituted furfural derivatives were used as substrates to demonstrate the generality of the presented methodology. In all cases, the corresponding 2-substituted piperidines were selectively formed in good yields (Fig. 5C and Supplementary Table 8).

In summary, we have developed the synthesis of piperidine from the important bio-based platform chemical furfural via direct

amination in the presence of a specific supported bimetallic catalyst. Under comparably mild conditions, the desired product can be obtained in yields up to 93% in the presence of the optimal Ru$_1$Co$_{NP}$/HAP system. Despite the complicated reaction network, excellent chemo- and regioselectivities are achieved for the individual steps of this cascade process and a clear mechanism is provided. Interestingly, lowering the concentration of the noble metal Ru on the support increases the activity of the catalyst system. This behaviour is explained by a surface single-atom alloy structure, which is only enabled at high Co/Ru ratios (e.g., Co/Ru ≥ 10) on the HAP support. The unique activity of the Ru$_1$Co$_{NP}$ SSAA catalyst was also extended to the synthesis of industrially important pyridine and substituted piperidines. As highlighted by the preparation of the actual pharmaceutical Artane, the presented methodology allows to access piperidine-based pharmaceuticals and agrochemicals from renewable biomass.

## Methods
### Catalyst preparation
All catalysts were prepared by incipient wetness impregnation. As an example, for the preparation of Ru$_1$Co$_{20}$/HAP catalyst, 209 mg Co(OAc)$_2$·4H$_2$O and 135 mg 3.17 wt$_{Ru}$% RuCl$_3$ were added to 1.4 g water and sonicated for 10 min, followed by the addition of 1 g HAP (calcined at 500 °C for 2 h). Then the mixture was evaporated under freeze-drying for 12 h. The obtained solid was ground to powder and then transferred to the tube furnace and then heated to 400 °C in air atmosphere at a ramp of 5 °C /min and was held at that temperature for 2 h. After cooling to room temperature, the gas was switched to hydrogen atmosphere and the tube was fluxed with hydrogen for 30 min. The tube furnace was then heated to 400 °C in hydrogen atmosphere at a ramp of 2 °C /min and was held at that temperature for 2 h. After being cooled to room temperature, the obtained sample

was defined as Ru$_1$Co$_{20}$/HAP catalyst and then transferred to the reaction mixture without being exposed to air.

## Catalytic reaction tests

In the typical reaction for the reductive amination of furfural, 0.5 mmol furfural (FAL, purified by bulb-to-bulb distillation under reduced pressure), 50 mg catalyst (without being exposed to air), and 5 g p-xylene (Alfa, anhydrous) were put into polytetrafluorethylene chamber in an autoclave (Parr reactor with a volume of 50 mL). After sealing the autoclave, the autoclave was purged with NH$_3$ for three times, and charged with 0.5 MPa NH$_3$ and 1 MPa H$_2$ at room temperature. Then the reaction mixture was stirred at a rate of 800 r/min and heated at 100 °C for 6 h subsequently heated for 20 min to 180 °C for 14 h. After the reaction, the liquid-phase products were analyzed with a GC system (Agilent 7890 A) equipped with a HP-5 column (30 m × 0.25 um × 0.25 mm i.d) and a FID detector by using dodecane as an internal standard.

The conversion of FAL ($X_{FAL}$) and the yield of piperidine ($Y_{piperidine}$) were calculated using the following equations:

$$X_{FAL}(\%) = \frac{mol_{FAL\ consumed}}{mol_{FAL\ fed}} \times 100 \tag{1}$$

$$Y_{piperidine}(\%) = \frac{mol_{piperidine\ produced}}{mol_{FAL\ fed}} \times 100 \tag{2}$$

**Overall piperidine production rate** in Table 1:

$$overall\ piperidine\ production\ rate(mol_{2a}\cdot mol_{Ru}^{-1}\cdot h^{-1}) = \frac{produced\ piperidine(mol)}{Ru\ loading\ (mol) \times 20\ (h)} \tag{3}$$

## Synthesis of pyridine

Following the general procedure, the reaction mixture was stirred under room temperature for 30 mins to release the NH$_3$. Then, the autoclave was further purged with N$_2$ for three times and charged with 2 MPa N$_2$ at room temperature. Then the reaction mixture was stirred at a rate of 800 r/min and heated at 240 °C for 24 h. After the reaction, the liquid-phase products were analyzed with a GC system (Agilent 7890 A) equipped with a HP-5 column (30 m × 0.25 um × 0.25 mm i.d) and a FID detector by using dodecane as an internal standard.

## Synthesis of piperidine hydrochloride

Following the general procedure, the Ru$_1$Co$_{20}$/HAP catalyst was removed by centrifugation from the reaction mixture. Then, 5 ml 2 M aqueous hydrochloric acid solution was added to the solution, and the piperidine was extracted in H$_2$O phase. The pure piperidine hydrochloride solid was obtained by evaporation.

## Synthesis of artane

0.5 mmol obtained piperidine hydrochloride, 0.5 mmol acetophenone, 1 mmol paraformaldehyde were added into 50 mL 0.1 M HCl ethanol solution and refluxed at 100 °C for 6 h. The mixture was evaporated and the obtained solid was washed by 50 mL diethyl ether, and then collected by filtration and alkalized with 1 M NaOH, finally to obtain β-piperidinopropiophenone. Then, β-piperidinopropiophenone and 2 mL dry THF were added in Schlenk tube under anhydrous and oxygen-free conditions in ice bath, the 2 M cyclohexylmagnesium chloride in ether was added drop by drop to the solution, to get the final product Artane.

**The actual Co and Ru loadings** were determined by inductively coupled plasma spectroscopy (ICP-OES) on an IRIS Intrepid II XSP instrument (Thermo Electron Corporation).

**H$_2$-TPR** was carried out with a Micromeritics AutoChem II 2920 System. 0.10 g of the calcined sample was loaded in a quartz reactor, heated in Ar flow at 300 °C for 1 h with a ramp of 10 °C/min, and then cooled down to 50 °C. The reactor was flushed with 10% H$_2$/Ar to reach a stable background. Then the sample was heated to 800 °C at a rate of 10 °C /min in 10 vol% H$_2$/Ar with a flow rate of 30 mL/min.

**The high-angle annual dark-filed scanning transmission electron microscopy (HAADF-STEM) and energy dispersive X-ray spectroscopy (EDS)** experiments were performed on a JEOL JEM-2100F microscope operated at 200 kV, equipped with an Oxford Instruments ISIS/INCA energy-dispersive X-ray spectroscopy (EDS) system with an Oxford Pentafet Ultrathin Window (UTW) Detector. Before microscopy examination, the sample was ultrasonically dispersed in ethanol for 15-20 min, and then a drop of the suspension was dropped on a copper TEM grid coated with a thin holey carbon film.

**In-situ X-ray diffraction (XRD)** analysis was carried out on a PANalytical X'pert diffractometer using Cu Kα radiation source (λ = 0.15432 nm) with a scanning angle (2θ) of 10°− 80°, operated at 40 kV and 40 mA. The Ru$_1$Co$_{20}$/HAP catalyst was transferred to the chamber with H$_2$ flow of 20 mL/min, and the XRD spectra were collected with elevating the temperature at every stage of 50 °C.

**In-situ X-ray photoelectron spectroscopy (XPS)** spectra were obtained on a Thermo ESCALAB 250 X-ray photoelectron spectrometer equipped with Al Kα excitation source and with C as internal standard (C 1 s = 284.6 eV). The Ru$_1$Co$_{20}$/HAP catalyst was transferred to the XPS chamber with 10 vol% H$_2$/Ar flow of 10 mL/min for 1 h, and the XPS spectra were collected after being cooled to room temperature.

**X-ray absorption spectra (XAS)** including X-ray absorption near edge structure (XANES) and extended X-ray absorption fine structure (EXAFS) at Ru K-edge of the samples were measured at the beamline 14 W of Shanghai Synchrotron Radiation Facility (SSRF) in China. The output beam was selected by Si (311) monochromator, and the energy was calibrated by Ru foil. The data were collected at room temperature under transmission mode. Athena software package was employed to process the XAS data. The samples were reduced at 400 °C for 2 h and directly sealed in the Kapton film without being exposed to air.

**NMR spectra** were recorded at room temperature in CDCl$_3$ on 400 MHz Bruker DRX-400 NMR spectrometers.

## Computational methods

All DFT calculations were carried out with the Vienna Ab-initio Simulation Package (VASP)[49]. The electron exchange and correlation energy was treated by generalized gradient approximation (GGA) based on the Perdew−Burke−Ernzerhof (PBE) functional[50]. Projector-augmented-wave (PAW) potential was employed to describe the interaction between ions and electronss[51]. The Kohn−Sham wave functions was expanded with a plane-wave basis set with a cut-off energy of 400 eV[52]. The thermodynamically Ru$_1$/Co (001) surface were used to simulate the Co surface of Ru$_1$Co$_{20}$/HAP catalyst[47]. The Ru$_1$/Co(001) were modeled with four-layer-thick slabs. The upper two layers of the slabs together with the adsorbates were allowed to relax, whereas the bottom two layers fixed at bulk position during the structure optimizations. A 15 Å of vacuum layer was used to separate the surface from the periodic image. The Brillouin zone was sampled by (3 × 3 × 1) Monkhorst−Pack k-point mesh[53]. The convergence criterion for the electronic self-consistent iteration and force were set to 10$^{-5}$ eV and 0.05 eV/Å, respectively. The adsorption energies ($E_{ads}$), were calculated by $E_{ads} = E_{adsorbate+surface} − (E_{adsorbate} + E_{surface})$, where $E_{adsorbate+surface}$, $E_{surface}$ and $E_{adsorbate}$ are total energy of surface covered with adsorbates, the energy of clean surface and the energy of free adsorbate, respectively.

## Data availability

The data that support the findings of this study are available from the corresponding author upon reasonable request.

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

## Acknowledgements

The authors are grateful for the supports from the National Natural Science Foundation of China (22132006, 21721004, 22172159, 22209171), CAS Project for Young Scientists in Basic Research (YSBR-022), the Youth Innovation Promotion Association CAS (2022185), and the Strategic Priority Research Program of the Chinese Academy of Sciences (XDA21060203). Haifeng Qi thanks the Alexander von Humboldt Foundation (CHN 1220532 HFST-P). This research used Beamline BL14W1 of the Shanghai Synchrotron Radiation Facility (SSRF).

## Author contributions

H.Q. synthesized the catalyst and performed most of the reactions and characterizations. Y.L., Y.C., and X.D. helped do the DFT calculations. Z.Z., F.L., W.G., L.Z., and K.J. helped the analysis with constructive discussions. X.L. helped analysis XAS data. L.L. helped analysis IR results. Y.S. helped do the STEM tests. K. J. gave the guidance for the substrate scope and revised manuscript. H.Q., Y.C., and A.W. wrote the manuscript. X.D., M.B., A.W., and T.Z. revised the paper. X.D., M.B., A.W., and T.Z. designed the study and supervised the project.

## Funding

## Competing interests

The authors declare no competing interests.
