## [Peer Review file · Nature Communications]

REVIEWER COMMENTS

Reviewer #1 (Remarks to the Author):

The current form of manuscript is substantially revised and updated by addressing the mentioned suggestions. Though, I am not fully convinced, in fact, surprised that why authors are avoiding even the basic characterization (even the PXRD of Ru₁Co₈₀/HAP and Ru₁Co₁₈₀/HAP is not provided) of the mentioned catalysts, which is essentially important to affix the primary form of the material (here catalyst). As author emphasized, I would also like to understand why it is a negative to have sufficiently isolated Ru atoms, particularly when Ru₁Co₈₀/HAP (4x) and Ru₁Co₁₈₀/HAP (9x) displayed several folds higher activity than Ru₁Co₂₀/HAP ?

After convincingly addressing the above queries this manuscript may be considered for publication.

Reviewer #2 (Remarks to the Author):

The authors have adequately responded to all of my previous concerns. I recommend the paper for publication.

Reviewer #3 (Remarks to the Author):

The authors have addressed most of my comments. Before it can be accepted, I still have one question/suggestion:

The characterization of Ru₁Co₂₀/HAP shows that few Ru-Ru exists in it. Based on the different loadings (Ru₁Co₄₀, Ru₁Co₈₀, Ru₁Co₁₆₀), the author concluded that the surface single-atom alloy Ru₁CoNP Catalyst plays a key role in this tandem reaction. However, no characterization results can prove that Ru₁Co₄₀, Ru₁Co₈₀, and Ru₁Co₁₆₀ are single-atom alloy catalysts. Is there any way (such as CO-FTIR) to

prove this though it is difficult as mentioned by the authors? If not, it would be better to rethink the conclusion as no strong evidence.

Responses to the referees

Reviewer #1 (Remarks to the Author):

1. The current form of manuscript is substantially revised and updated by addressing the mentioned suggestions.

Response: Thanks for your positive comment.

2. Though, I am not fully convinced, in fact, surprised that why authors are avoiding even the basic characterization (even the PXRD of Ru₁Co₈₀/HAP and Ru₁Co₁₈₀/HAP is not provided) of the mentioned catalysts, which is essentially important to affix the primary form of the material (here catalyst).

Response: Previously, in order to obtain a direct evidence of single-atom Ru species, we have already done the basic characterizations of PXRD (Supplementary Fig. 3 in SI), CO-DRIFTS (Supplementary Fig. 6 in SI), aberration-corrected high-angle annular dark-field scanning transmission electron microscopy (AC-HAADF-STEM, Fig. R1a) and *in-situ* H₂-reduction environmental TEM (ETEM, Fig. R1b) for Ru₁Co₂₀/HAP sample, but no obvious Ru atomic signal were identified. So, we thought it would be also difficult to obtain effective structural information even characterizing Ru₁Co_y/HAP (y=40, 80, and 160) materials with lower Ru content, that's the reason why we didn't further perform the basic characterizations for these low-Ru-loading catalysts. In fact, according to reviewer's suggestion for last version of manuscript, we have already performed the PXRD characterization for Ru₁Co₄₀/HAP catalyst (see in Supplementary Fig. 3 in SI), indeed neither Ru signal nor other effective information can be obtained. Fortunately, at this stage we have obtained the reliable XAS data for the Ru₁Co₄₀/HAP and Ru₁Co₈₀/HAP samples (Fig. R2 and Table R1), which proves the single-atom Ru dispersion.

Figure R1. (a) AC-HAADF-STEM and (b) *in-situ* H₂-reduction ETEM images of Ru₁Co₂₀/HAP. (*In-situ* H₂-reduction process: the ETEM images were collected after H₂ reduction under 400 °C for 2h in TEM chamber).

In order to further prove our conclusion “a unique surface single-atom alloy (SAA) structure enables the one-pot formation of piperidine from furfural”, the Ru₁Co₄₀/HAP and Ru₁Co₈₀/HAP catalysts with relatively lower Ru content were chosen for X-ray absorption spectra (XAS) characterizations. This time, a long-time XAS signal acquisition was applied for trying to obtain high-quality XAS data. As shown in Fig. R2a-b, the obvious different *k*-space patterns of the Ru₁Co₄₀/HAP and Ru₁Co₈₀/HAP with that of Ru foil as well as only one peak at ~1.98 Å (not phase-corrected) in *r*-space of these two catalysts, suggest the formation of sole Ru-Co coordination, which is also confirmed by wavelet transforms (Fig. R2c). The best-fitted EXAFS results of the Ru₁Co₄₀/HAP and Ru₁Co₈₀/HAP samples reveal Ru-Co shell at 2.46 Å with coordination number (CN) of 5.3 and Ru-Co shell at 2.48 Å with CN of 5.7, respectively (Fig. R2e-f and Table R1). The absence of Ru-Ru shell undoubtedly indicates the single-Ru-atom dispersion on Co nanoparticles in the Ru₁Co₄₀/HAP and Ru₁Co₈₀/HAP catalysts.

(Note: High-energy XAS experiment, e.g., for Ru XAS characterization, is usually open during July-August and November-December in Shanghai Synchrotron Radiation Facility (SSRF) every year. Therefore, we can not guarantee the XAS measurement for Ru element at any time of the year. Last year, we performed the XAS characterizations

for the Ru₁Co₂₀/HAP and Ru₁Co₄₀/HAP catalysts, but only a reliable signal of Ru₁Co₂₀/HAP sample was obtained. According to reviewers' suggestions, in July this year, again we tried the XAS characterizations for the low-Ru-loading Ru₁Co₄₀/HAP and Ru₁Co₈₀/HAP samples, but a long-time XAS signal acquisition was applied. Fortunately, moderate-quality XAS data of both Ru₁Co₄₀/HAP and Ru₁Co₈₀/HAP samples were obtained this time, seen in Fig. R2).

Figure R2. The k^2 -weighted Fourier transform extended X-ray absorption fine structure spectra (FT-EXAFS) in k -space (a) and r -space (b) as well as wavelet transforms (c), and the corresponding FT-EXAFS fitting curves of Ru₁Co₂₀/HAP (d), Ru₁Co₄₀/HAP (e), and Ru₁Co₈₀/HAP (f).

Table R1. The best-fitted EXAFS results of Ru₁Co₄₀/HAP and Ru₁Co₈₀/HAP catalysts.^a

Sample	Shell	CN	R (Å)	σ^2 (10^{-2} Å ²)	ΔE_0 (eV)	r-factor (%)
Ru foil	Ru-Ru	12	2.67	0.3	0.6	0.8
Ru ₁ Co ₄₀ /HAP	Ru-Co	5.3	2.46	0.6	-19.2	0.8
Ru ₁ Co ₈₀ /HAP	Ru-Co	5.7	2.48	0.6	-15.4	1.1

^aCN is the coordination number for the absorber-backscatterer pair, R is the average absorber-backscatterer distance, σ^2 is the Debye-Waller factor, and ΔE_0 is the inner potential correction. The accuracies of the above parameters are estimated as CN, $\pm 20\%$; R, $\pm 1\%$; σ^2 , $\pm 20\%$; ΔE_0 , $\pm 20\%$. The data range used for data fitting in k -space (Δk) and R -space (ΔR) are 3.0-11.3 Å⁻¹ and 1.0-2.6 Å, respectively.

3. As author emphasized, I would also like to understand why it is a negative to have sufficiently isolated Ru atoms, particularly when Ru₁Co₈₀/HAP (4x) and Ru₁Co₁₈₀/HAP (9x) displayed several folds higher activity than Ru₁Co₂₀/HAP?

Response: We are sorry for the misunderstanding of the negativity for having sufficiently isolated Ru atoms, here the Ru content in Ru₁Co₄₀/HAP (0.21 wt% Ru), Ru₁Co₈₀/HAP (0.10 wt% Ru), and Ru₁Co₁₆₀/HAP (0.05 wt% Ru, see Table 1 in manuscript) decreases gradually but the Co loading remains constant (5 wt% Co). Because of the complicated two-step tandem catalysis (100 °C for 6 h: reductive amination of furfural, and 180 °C for 14 h: hydrogenative ring arrangement of furfurylamine to piperidine), so we calculated the overall piperidine production rate based on total Ru content and whole reaction time: overall piperidine production rate ($mol_{2a} \cdot mol_{Ru}^{-1} \cdot h^{-1}$) = $\frac{\text{produced piperidine (mol)}}{\text{Ru loading (mol)} \times 20 \text{ (h)}}$ (already been supplemented in the manuscript), and just wanted to show that only at very dilute concentrations of Ru piperidine formation occurred. So, we can not conclude that it is negative to have sufficiently isolated Ru atoms based on the piperidine production rate.

Actually, it is positive for reactivity to have isolated Ru atoms: the higher the isolated Ru atomic ratio, the higher the selectivity.

Action taken: We have already supplemented the XAS data of the Ru₁Co₄₀/HAP and Ru₁Co₈₀/HAP catalysts in Fig. 3, Supplementary Fig. 4 and Table 3 in the revised manuscript, corresponding explanation is seen on page 10/upper, “In order to prove our hypothesis, the Ru₁Co₄₀/HAP and Ru₁Co₈₀/HAP catalysts with relatively lower Ru content were further chosen to perform XAS characterizations. As shown in Fig. 3c-d, the remarkably different *k*-space patterns of the Ru₁Co₄₀/HAP and Ru₁Co₈₀/HAP with that of Ru foil as well as only one peak at ~1.98 Å (not phase-corrected) in *r*-space of these two catalysts, suggest the formation of sole Ru-Co coordination, which is also confirmed by wavelet transforms (Fig. 3e). The best-fitted EXAFS results of the Ru₁Co₄₀/HAP and Ru₁Co₈₀/HAP samples reveal Ru-Co shell at 2.46 Å with CN of 5.3 and Ru-Co shell at 2.48 Å with CN of 5.7, respectively (Supplementary Fig. 4b-c and Supplementary Table 3). The absence of Ru-Ru coordination clearly indicates the isolated Ru atoms anchored on Co nanoparticles in the low-Ru-loading Ru₁Co_y/HAP samples (*y* = 40, 80, and 160)”.

And we have also supplemented the calculation method for overall piperidine production rate in the manuscript, seen on page 13/down, “Overall piperidine production rate in table 1: overall piperidine production rate ($mol_{2a} \cdot mol_{Ru}^{-1} \cdot h^{-1}$) = $\frac{\text{produced piperidine (mol),,}}{\text{Ru loading (mol)} \times 20 \text{ (h)}}$ ”.

4. After convincingly addressing the above queries this manuscript may be considered for publication.

Response: We hope this version of manuscript is acceptable for publication in Nature Communications.

Reviewer #2 (Remarks to the Author):

1. The authors have adequately responded to all of my previous concerns. I recommend the paper for publication.

Response: Thanks for your positive comment.

Reviewer #3 (Remarks to the Author):

1. The authors have addressed most of my comments.

Response: Thanks for your positive comments.

2. Before it can be accepted, I still have one question/suggestion: The characterization of Ru₁Co₂₀/HAP shows that few Ru-Ru exists in it. Based on the different loadings (Ru₁Co₄₀, Ru₁Co₈₀, Ru₁Co₁₆₀), the author concluded that the surface single-atom alloy Ru₁CoNP Catalyst plays a key role in this tandem reaction. However, no characterization results can prove that Ru₁Co₄₀, Ru₁Co₈₀, and Ru₁Co₁₆₀ are single-atom alloy catalysts. Is there any way (such as CO-FTIR) to prove this though it is difficult as mentioned by the authors? If not, it would be better to rethink the conclusion as no strong evidence.

Response: Same as responded in question 2 by referee 1, we have tried the XAS characterizations again for low-Ru-loading Ru₁Co₄₀/HAP and Ru₁Co₈₀/HAP catalysts by long-time signal acquisition, and already obtained reliable XAS data (Fig. R2 and Table R1), which strongly proves Ru species are sufficiently isolated by the neighbouring Co atoms and dispersed as single atoms in these low-Ru-loading catalysts.